# Capsaicin Reduces Obesity by Reducing Chronic Low-Grade Inflammation

**DOI:** 10.3390/ijms25168979

**Published:** 2024-08-18

**Authors:** Jiaxin Yang, Wanyi Li, Yuanwei Wang

**Affiliations:** College of Animal and Veterinary Sciences, Southwest Minzu University, Chengdu 610041, China; 220952002050@stu.swun.edu.cn (J.Y.); 230952002050@stu.swun.edu.cn (W.L.)

**Keywords:** capsaicin, chronic low-grade inflammation, anti-obesity, gut barrier, gut microbiota

## Abstract

Chronic low-grade inflammation (CLGI) is associated with obesity and is one of its pathogenetic mechanisms. Lipopolysaccharide (LPS), a component of Gram-negative bacterial cell walls, is the principal cause of CLGI. Studies have found that capsaicin significantly reduces the relative abundance of LPS-producing bacteria. In the present study, *TRPV1*-knockout (*TRPV1*^−/−^) C57BL/6J mice and the intestinal epithelial cell line Caco-2 (*TRPV1*^−/−^) were used as models to determine the effect of capsaicin on CLGI and elucidate the mechanism by which it mediates weight loss in vivo and in vitro. We found that the intragastric administration of capsaicin significantly blunted increases in body weight, food intake, blood lipid, and blood glucose in *TRPV1*^−/−^ mice fed a high-fat diet, suggesting an anti-obesity effect of capsaicin. Capsaicin reduced LPS levels in the intestine by reducing the relative abundance of Proteobacteria such as *Helicobacter*, *Desulfovibrio,* and *Sutterella*. Toll-like receptor 4 (TLR4) levels decreased following decreases in LPS levels. Then, the local inflammation of the intestine was reduced by reducing the expression of tumor necrosis factor (TNF)-α and interleukin (IL)-6 mediated by TLR4. Attenuating local intestinal inflammation led to the increased expression of tight junction proteins zonula occludens 1 (ZO-1) and occludin and the restoration of the intestinal barrier function. Capsaicin increased the expression of ZO-1 and occludin at the transcriptional and translational levels, thereby increasing trans-endothelial electrical resistance and restoring intestinal barrier function. The restoration of intestinal barrier function decreases intestinal permeability, which reduces the concentration of LPS entering the circulation, and reduced endotoxemia leads to decreased serum concentrations of inflammatory cytokines such as TNF-α and IL-6, thereby attenuating CLGI. This study sheds light on the anti-obesity effect of capsaicin and its mechanism by reducing CLGI, increasing our understanding of the anti-obesity effects of capsaicin. It has been confirmed that capsaicin can stimulate the expression of intestinal transmembrane protein ZO-1 and cytoplasmic protein occludin, increase the trans-epithelial electrical resistance value, and repair intestinal barrier function.

## 1. Introduction

Obesity has become one of the most significant health problems worldwide; its incidence worldwide is increasing yearly [1]. Various chronic diseases induced by obesity threaten public health and impose heavy economic burdens on society [2]. Obesity is influenced by various factors, including traditional ones such as energy balance and genetic predispositions and environmental factors like dietary and exercise habits. Recent studies have also linked obesity to molecular-level factors, including alterations in gut microbiota, short-chain fatty acids, neurohumoral regulatory systems, localized inflammation in intestinal or adipose tissue, and systemic inflammatory responses [3]. Several lines of evidence suggest that chronic low-grade inflammation (CLGI) is associated with obesity [4,5]. Lipopolysaccharide (LPS) is a component of the cell wall of Gram-negative bacteria and is the main pro-inflammatory component of bacteria. Studies have reported that obesity and related diseases caused by Western high-calorie diets might be caused by increased systemic LPS concentrations (i.e., endotoxemia) [6,7]. LPS-induced endotoxemia has been demonstrated to be the primary mechanism of CLGI [8]. These findings suggest that increased numbers of Gram-negative bacteria in the intestine will cause CLGI and lead to obesity. Proteobacteria is the largest phylum of bacteria, and all bacteria in the phylum are Gram-negative. The relative abundance of Proteobacteria in the gastrointestinal tract of healthy people is between 2% and 5%; in individuals with obesity, metabolic disorder, or intestinal inflammation, the relative abundance is as high as 15% [9]. As a component of the Gram-negative bacterial cell wall, the concentration of LPS increases with the relative abundance of Proteobacteria. Studies have shown that the increased abundance of Proteobacteria in the intestinal flora activates immune signaling pathways, leading to CLGI [10], suggesting that the abundance of Proteobacteria in the intestine may substantially impact metabolism and may be a risk factor for obesity. Previous studies found a high-fat diet-induced increase in the number of *Escherichia*, *Helicobacter*, *Desulfovibrio*, and *Sutterella* bacteria in Proteobacteria in the intestinal of mice, while capsaicin significantly reduced their relative abundance [11]. For these reasons, we hypothesized that capsaicin would mediate weight loss by reducing CLGI.

The intestinal barrier is critically involved in the development of CLGI. In a study of capsaicin reducing CLGI function, Kang et al. found that capsaicin significantly increased the mRNA expression of *occludens 1 (ZO-1)* and *occludin* gene [12]. Wang et al. studied the effects of Galactooligosaccharide on repairing intestinal barrier function and promoting the expression of *occludin*, *claudin-1*, and *ZO-1* gene [13]. Wang et al. found that paeoniflorin inhibited the ability of LPS to damage intestinal barrier function and increased the expression of *occludin*, *claudin-5*, and *ZO-1* gene to mediate repair of the intestinal barrier [14]. In a study of the effect of *Bifidobacteria* on intestinal function, Zhao et al. obtained similar results [15]. Nevertheless, the ability of capsaicin to repair intestinal barrier function requires more experimental support. We performed the present study to investigate whether repair may be one of the mechanisms by which capsaicin reduces CLGI and achieves weight loss.

Capsaicin, a vanillamide alkaloid derived from chili peppers, is characterized by a strong irritant odor [16]. Modern pharmacological studies have demonstrated that capsaicin possesses multiple therapeutic effects, including anti-itch and analgesic properties, weight loss and lipid regulation, anticancer activities, antibacterial effects, blood pressure reduction, endocrine system regulation, and the protection of cardiovascular, cerebrovascular, and digestive systems. Consequently, capsaicin has potential applications in treating conditions such as neuropathic pain, arthritis, obesity, diabetes, and cancer [17]. Capsaicin has an anti-obesity effect, and the mechanisms of this effect, which are mainly through activation of the transient receptor potential vanilloid-1 (TRPV1) channel, have been extensively studied in both rodents and other species [18]. Our previous study found that capsaicin still has an anti-obesity effect without TRPV1 cation channel activation. Therefore, the anti-obesity effect of capsaicin may involve an alternative mechanism, needing further studies. At the same time, the effect trend of capsaicin on the composition of the gut microbiota of C57BL/6J mice and *TRPV1*-knockout (*TRPV1*^−/−^) C57BL/6J mice was also consistent. This study aims to verify the effects of capsaicin on CLGI both in vivo and in vitro and to elucidate the mechanism by which capsaicin mediates weight loss independently of TRPV1 cationic channel activation, thereby further enriching the understanding of capsaicin’s weight-loss effects.

## 2. Results

### 2.1. Effects of Capsaicin on Body Weight and Food Intake

As shown in Table 1, after 12 weeks, the weight of mice in the HFD group (feed with high-fat diet) increased by 19.16 ± 0.63 g, significantly higher than that of the mice in the SLD group (feed with standard lipid diet) (*p* < 0.05). The weight of mice fed with HFD increased significantly, indicating that the obesity model was successfully established. Capsaicin significantly inhibited HFD-induced weight gain (*p* < 0.05). Body weights in the HFDC group (fed with a high-fat diet and capsaicin) increased by 13.43 ± 1.52 g, significantly lower than that of the HFD group, suggesting that capsaicin inhibited weight gain. The average weekly food intake was lower in the HFDC group than in the HFD group. After 12 weeks, the total food intake of the HFDC group was significantly lower than that of the HFD group (*p* < 0.05), suggesting that capsaicin inhibited food intake.

### 2.2. Effect of Capsaicin on Serum Biochemical Indices

The effects of capsaicin on serum biochemical indicators are shown in Figure 1. Levels of fasting blood glucose, triglycerides, total cholesterol, low-density cholesterol, and insulin in the HFD group were significantly higher than those in the SLD group (*p* < 0.05). Capsaicin significantly inhibited the HFD-induced increases in blood lipids and glucose. Compared with the HFD group, the mice of the HFDC group had 30% lower fasting blood glucose levels and 20% lower insulin levels. Triglyceride and cholesterol levels were also significantly lower than those of the HFD group (*p* < 0.05), suggesting that capsaicin has a lipid-lowering effect. There were no significant differences in triglyceride, total cholesterol, or low-density lipoprotein cholesterol levels between the HFDC and SLD groups (*p* > 0.05).

### 2.3. Effect of Capsaicin on Oral Glucose Tolerance in Mice

The oral glucose tolerance test was performed in the 11th week. Glucose tolerance was obtained by calculating the area under the curve (AUC) generated by blood glucose levels from 0 to 120 min. In the HFD group, which was fed a high-fat diet, blood glucose levels were higher than those of SLD mice between 15 and 120 min (Figure 2A). Capsaicin significantly inhibited the HFD-induced increase in AUC (Figure 2B), and the AUC of the HFDC group was 14% lower than the HFD group (*p* < 0.05), suggesting that capsaicin significantly reduced blood glucose levels in obese mice.

### 2.4. Effect of Capsaicin on Proteobacteria

The relative abundances of *Helicobacter*, *Desulfovibrio,* and *Sutterella* in fecal samples of the mice in each group were quantified using quantitative fluorescence PCR (Figure 3). These relative abundances in the HFD group were significantly higher than those of the SLD group (*p* < 0.05), and capsaicin significantly reduced these abundances, suggesting that capsaicin decreased abundance of Proteobacteria induced by HFD.

### 2.5. Changes in LPS Levels in the Intestine

Toll-like receptor 4(TLR4) is a specific receptor for LPS. *TLR4* gene expression increases with the increase in LPS concentration in the intestine. The results of mRNA expression of *TLR4* gene in intestinal tissue are shown in Figure 4. mRNA expression of *TLR4* gene in the intestinal tract of mice in the HFD group was significantly higher than that of the SLD group, suggesting that the HFD led to an increase in the number of LPS-producing bacteria and increased the concentrations of LPS in the intestine. The mRNA expression of *TLR4* gene increased correspondingly. The mRNA expression of *TLR4* gene in the HFDC group was significantly lower than that of the HFD group, suggesting that capsaicin reduced the relative abundance of LPS-producing bacteria in the intestine, thereby reducing the production of LPS and the mRNA expression of *TLR4* gene.

### 2.6. Effect of Capsaicin on Local LPS-Induced Inflammatory Responses

The effect of capsaicin on local inflammation induced by LPS in vivo was evaluated by measuring the mRNA expression and concentrations of tumor necrosis factor (TNF)-α and interleukin (IL)-6 in intestinal tissues and histopathological sections. As shown in Figure 5, pathological sections of intestinal tissue showed that the villus lengths of the duodenum, jejunum, and ileum of mice in the HFD group became shorter, and the recess depth became deeper, resulting in significantly lower villus gland ratios than in the SLD group (*p* < 0.05) (Figure 6). As shown in Figure 7, the mRNA expression of *TNF-α* and *IL-6* gene in the intestine of the HFD group was significantly higher than that of the control group (*p* < 0.05), and their concentrations in tissues were also significantly higher (*p* < 0.05), suggesting that HFD aggravates local intestinal inflammation. Compared with the HFD group, the mRNA expression of *TNF-α* and *IL-6* gene and their concentrations in intestinal tissue of the HFDC group were significantly lower (*p* < 0.05). Histopathology showed that intestinal lesions in the HFDC group were milder than those in the HFD group. The villus gland ratio was significantly greater (*p* < 0.05), suggesting that the local intestinal inflammatory response in the intestine of the mice was attenuated after feeding with capsaicin.

The effect of capsaicin on LPS-induced local inflammation in vitro was evaluated by measuring the mRNA expression and concentrations of TLR4, TNF-α, and IL-6 in Caco-2 (*TRPV1*^−/−^) cells. As shown in Figure 8, in vitro tests revealed that there was no significant difference in mRNA expression of *TLR4*, *TNF-α*, or *IL-6* gene in the cells of LPS and LPSC groups (*p* > 0.05), and there were no significant differences levels of *TNF-α* or *IL-6* gene (*p* > 0.05) in the supernatants of cell culture medium. However, the mRNA expression of *TLR4*, *TNF-α*, and *IL-6* gene and the concentration in the supernatants of the two groups were significantly higher than those of the control group (*p* < 0.05), suggesting that LPS caused local inflammatory responses in the Caco-2 (*TRPV1*^−/−^) cell model and that the inflammatory response model was established. Capsaicin did not affect the local inflammatory response in the Caco-2 (*TRPV1*^−/−^) cells.

### 2.7. Effect of Capsaicin on Intestinal Barrier Function

Transmembrane protein occludin and cytoplasmic protein ZO-1 are essential for maintaining the barrier function of epithelia. The detrimental impact of LPS on the intestinal barrier function has been well established. To investigate the effect of capsaicin on this function, an in vitro intestinal epithelial cell layer formed by Caco-2 (*TRPV1*^−/−^) cells was treated with LPS, and the integrity of the Caco-2 (*TRPV1*^−/−^) cell monolayer was assessed using the Trans-epithelial electrical resistance (TEER) method. The TEER values decreased significantly after 24 h of LPS treatment. Therefore, to determine whether the repair effect of capsaicin on the intestinal barrier was related to occludin and ZO-1, we measured mRNA and protein levels of occludin and ZO-1 in cells in vitro and small intestine tissues in vivo. As shown in Figure 9 and Figure 10, in intestinal tissues and cells, respectively, mRNA and protein levels of occludin and ZO-1 in the HFD and LPS groups (respectively) were significantly lower than those of the SLD or control group, suggesting that the damage of LPS to intestinal barrier function was achieved by destroying occludin and ZO-1. The mRNA and protein expressions of occludin and ZO-1 in the HFDC and LPSC groups treated with capsaicin were significantly higher than those in the HFD and LPS groups, suggesting that capsaicin repairs the intestinal barrier function by upregulating the expression of occludin and ZO-1.

LPS-mediated damage to intestinal barrier function has been confirmed. To study the effect of capsaicin on intestinal barrier function, intestinal epithelial cell layers formed by Caco-2 (*TRPV1*^−/−^) cells in vitro were treated with LPS. To evaluate the cell barrier effect, the monolayer integrity of Caco-2 (*TRPV1*^−/−^) cells was determined using the TEER method. TEER was measured at 3, 6, 12, 18, and 24 h after adding LPS and capsaicin. As shown in Figure 11, the TEER value of the LPS group decreased over time, and there was a significant decrease after 6 h (*p* < 0.05) to nearly 50% after 24 h, suggesting a destructive effect of LPS on the cell layer. Capsaicin significantly inhibited the effect of LPS. The TEER value of the LPSC group decreased less than that of the LPS group. After 12 h, there was a significant difference between the LPS and LPSC groups (*p* < 0.05). After 24 h, the TEER value of LPS groups was nearly 50% less than that of the LPSC group, suggesting that capsaicin reduced intestinal epithelial permeability and repaired the intestinal barrier.

### 2.8. Effects of Capsaicin on Systemic Endotoxemia and Inflammatory Response

The damage to intestinal barrier function leads to LPS entering the systemic circulation, resulting in endotoxemia, which leads to systemic inflammation. To study the effects of capsaicin on endotoxemia and systemic inflammation, we measured serum levels of LPS, TNF-α, and IL-6. As shown in Figure 12, serum levels of LPS, TNF- α, and IL-6 in the HFD group were significantly higher than those of the SLD group, suggesting that the HFD led to an increase in the number of LPS-producing bacteria in the intestine, thereby causing an increase in LPS level. In the HFDC group, the concentrations of LPS, TNF-α, and IL-6 were significantly lower than those in the HFD group, suggesting that capsaicin reduced the relative abundance of LPS-producing bacteria in the intestine, attenuating the damage to the intestinal barrier. The concentration of LPS entering the circulation was ultimately reduced, secondary endotoxemia was inhibited, and CLGI was ultimately reduced.

## 3. Discussion

We showed that capsaicin significantly reduced body weight, food intake, triglyceride, cholesterol, fasting blood glucose, glucose tolerance, and insulin levels in *TRPV1*^−/−^ mice fed an HFD. These results agree with our previous study [11], confirming that capsaicin lowers lipid levels without TRPV1 cation channel activation. We also quantified Proteobacteria in fecal samples using fluorescent PCR methods. We found that the relative abundances of *Helicobacter*, *Desulfovibrio*, and *Sutterella* in the HFDC group were significantly lower than in the HFD group, and the capsaicin-induced trend of Proteobacteria in the intestine also agreed with our previous results [11]. These findings suggest that capsaicin reduces the abundance of Proteobacteria.

CLGI is a significant cause of obesity [4,5], and LPS endotoxemia is a primary mechanism of CLGI. The increase in LPS concentration in the intestine leads to local inflammation of intestinal epithelial cells, damaging the intestinal barrier function. LPS then passes through the intestinal barrier and enters the systemic circulation, causing endotoxemia. Endotoxemia triggers the innate immune system to release large amounts of inflammatory cytokines, resulting in CLGI throughout the body, which ultimately contributes to obesity, an inflammatory response in itself [8].

TLR4 is a pattern-recognition receptor for LPS. The binding of LPS to TLR4 results in the release of inflammatory cytokines and local intestinal inflammation [19]. Increased LPS concentrations in the intestines of obesity-prone rats coincided with the increase in TLR4 expression. However, TLR4 was not expressed in the intestine of non-obese rats [8]. In the present study, the mRNA expression of *TLR4* gene in the intestinal tract of the HFDC group was significantly lower than that of the HFD group, and the expression of *TLR4* gene positively correlated with LPS levels, suggesting that the concentration of LPS in the intestinal tract of the HFDC group decreased significantly and was consistent with the quantitative detection of Proteobacteria. Bacteria of the phylum Proteobacteria are Gram-negative and produce large amounts of LPS in the intestine. The decrease in its relative abundance likely results in decreased LPS concentrations in the intestine.

LPS plays an essential role in intestinal and systemic inflammation [20]. Jamar et al. found that the concentration of LPS increased, and TLR4 in the intestine was activated in mice fed with Western-style HFD, leading to the release of inflammatory cytokines (TNF-α, IL-6, IL-10, and IL-1 β) and ultimately to local intestinal inflammation [21]. In the present study, mRNA expression levels and concentrations of TNF-α and IL-6 in the intestinal tissue of the HFDC group were significantly lower than those of the HFD group. The pathological changes in the intestinal tissue in the HFDC group were also less than those in the HFD group, suggesting that capsaicin reduced local intestinal inflammation in the HFDC group. We found that the mRNA expression levels of *TNF-α* and *IL-6* gene and their concentrations in intestinal tissue of mice in the HFD group were significantly higher than those of the SLD group, suggesting that HFD leads to an increase in the number of LPS-producing bacteria in the intestine, thereby causing increased LPS concentrations and local inflammation. Our findings suggest that capsaicin reduces the concentration of LPS in the intestine by reducing the number of LPS-producing bacteria and weakening the local inflammatory response.

Other studies showed that capsaicin inhibited the LPS-induced release of inflammatory cytokines, including TNF-α, IL-6, IL-10, and IL-1 β [22,23,24]. Tang et al. found that capsaicin inhibited LPS-induced inflammatory cytokine release by upregulating the human liver X receptor Alpha [25]. Takeshi et al. found that capsaicin treated chronic gastritis induced by *Helicobacter* in Mongolian gerbils [26]. The study of the mechanism found that the TRPV1 receptor plays a substantial role [24,27,28,29,30]. In contrast, in vitro experiments showed that the mRNA expression of *TLR4*, *TNF-α*, and *IL-6* gene in the cells of the LPSC group was not significantly different from those of the LPS group, and levels of TNF-α and IL-6 in cell supernatants were not significantly different from those of the LPS group but were significantly higher than those of the control group. These findings suggest that the inhibition of local inflammation by capsaicin in intestinal tissue, as in other tissue, depends on the TRPV1 receptor. The cells used in our study were *TRPV1*-deletion strains, which explains why the in vitro experiments did not appear to demonstrate inhibition of inflammatory cytokines. We found that the mRNA expression of *TLR4*, *TNF-α*, and *IL-6* gene and the concentrations of TNF-α and IL-6 in the intestinal tissue of the HFDC group of *TRPV1*-deletion mice were significantly lower than those of the HFD group, which was attributed to the reduction in the number of LPS-producing bacteria in the intestinal tract by capsaicin, thereby reducing the local inflammatory response caused by LPS. The inhibitory effect of capsaicin on inflammatory factors through the TRPV1 receptor was not involved.

Intestinal epithelial cells (IECs) form tight junctions (TJs) at the apical side of the intestinal epithelium. TJ proteins are the most critical intercellular junctions in the junctional multiprotein complex [31]. These proteins seal off paracellular gaps between adjacent IECs to form a selectively permeable intercellular barrier that regulates molecular paracellular movement between the intestinal lumen and the subepithelial tissue. The formation of functional TJs is essential for maintaining intestinal permeability and intestinal barrier function [32,33]. Physiological functions of the TJ are maintained by ZO-1, occludin, and claudins. Studies have shown that the upregulation of ZO-1 and occludin inhibits increases in intestinal permeability [34]. The prevention of the destruction of ZO-1 and occludin is essential for the maintenance of intestinal barrier function [35,36]. Local inflammation increases intestinal permeability by destroying the TJs formed by the interaction between the transmembrane protein occludin and the cytoplasmic protein ZO-1 [6,7]. The present study showed that the mRNA and protein expressions of *ZO-1* and *occludin* gene were significantly higher in the HFDC group than the HFD group but lower than in the SLD group. This finding suggests that the HFDC group’s barrier function is less impaired than that of the HFD group, which is consistent with the results of Kang et al. and Santos et al. concerning the repairment of the intestinal barrier function by capsaicin [12,37]. Our findings suggest that capsaicin reduces the concentration of LPS in the intestine by reducing the number of LPS-producing bacteria, thereby reducing the damage to the intestinal barrier function while increasing the mRNA and protein expression levels of ZO-1 and occludin. The same results were obtained in a study of substances that repair the intestinal barrier function, including *Bifidobacterium*, paeoniflorin, and Galactooligosaccharide [13,14,15]. Our in vitro experiments showed that, at the same concentration of LPS, the mRNA and protein expressions of ZO-1 and occludin in the LPSC group supplemented with capsaicin were significantly higher than those of the control group, suggesting that capsaicin mediates the repair of intestinal barrier function. This may be attributed to capsaicin’s ability to repair intestinal barrier function, thereby reducing lipid absorption in the intestine and causing blood lipid levels in the HFDC group to return to normal levels similar to those in the SLD group.

TEER measures the information of the ion current resistance of monolayer cells, which is directly related to the integrity of intercellular TJs [38]. Higher TEER values suggest more complete monolayers and lower permeability, representing the integrity of the barrier function. In the present study, we measured the restoration of the intestinal barrier function by capsaicin by measuring TEER. We found that at the same concentration of LPS, the TEER value of the LPS group decreased significantly, suggesting that the model can be used as the control group. The TEER value of the LPSC group was significantly higher than that of the LPS group, and the permeability decreased, further suggesting the reparative effect of capsaicin on intestinal barrier function. This result is consistent with the decrease in the LPS concentration in blood and the attenuation of endotoxemia in vivo.

The impairment of intestinal barrier function leads to the entry of LPS into the systemic circulation through the damaged intestinal epithelium, i.e., endotoxemia [7,21]. Endotoxemia activates the innate immune system to synthesize and release large amounts of inflammatory cytokines [39]. The levels of LPS, TNF-α, and IL-6 in the blood of mice in the HFD group were significantly higher than those in the SLD group, while levels in the HFDC group were significantly lower than those in the HFD group but higher than those of the SLD group. This finding suggests that capsaicin reduces LPS levels, endotoxemia, and, therefore, systemic CLGI. Although the result is consistent with the conclusion of Kang et al., they showed that bacteria from the S24-7 family were responsible for increasing the LPS concentration in mice fed an HFD, which induced CLGI [12], while we showed that Proteobacteria cause CLGI. S24-7 are members of the phylum Bacteroidetes; a higher relative abundance of the Bacteroides may benefit obese people [40]. Other studies showed that S24-7 has therapeutic effects on glucose intolerance and obesity [41,42]. CLGI caused by the phylum Proteobacteria has also been demonstrated [43,44,45].

## 4. Materials and Methods

### 4.1. Materials

Capsaicin (M2028, ≥95%) and LPS (L2880) were purchased from Sigma-Aldrich, Inc. (St. Louis, MO, USA). A standard lipid diet (SLD, D12450B, 10% calories as fat) and a high-fat diet (HFD, D12492, 60% calories as fat) were purchased from Research Diets (New Brunswick, NJ, USA). All other reagents were of the highest commercially available grade.

### 4.2. Animals and Experimental Design

Eight-week-old female B6.129X1-Trpv1tm1Jul/J (*TRPV1*^−/−^) mice were housed under standard conditions (temperature, 22 ± 2 °C; humidity, 55 ± 5%), with free access to food and water and a 12 h light/dark cycle. After a 1-week acclimation and feeding of an SLD for 2 weeks, *TRPV1*^−/−^ mice were randomly divided into three groups (n = 10 each): the SLD group (feed with SLD), the HFD group (feed with HFD), and the HFDC group (feed with HFD and capsaicin), with one mouse per cage. Each mouse in the HFDC group was intragastrically fed capsaicin 2 mg/kg body mass capsaicin dissolved in 0.9% saline containing 3% ethanol and 10% Tween-80 on alternate days, and SLD and HFD group mice were given the corresponding vehicle for 12 weeks.

The three days before the end of the experiment, fresh feces from each mouse were collected and mixed for three consecutive days, then stored at −80 °C until use. The day before the end of the experiment, the mice in each group were fasted for 12 h and then sacrificed after CO_2_ anesthesia. Fresh blood was anticoagulated with EDTA and subsequently used for serum preparation. The serum was packaged and freezing at −80 °C. Two centimeters of each mouse’s duodenum, jejunum, and ileum were collected and stored in liquid nitrogen.

The human colon adenocarcinoma cell line Caco-2 (*TRPV1*^−/−^) with TRPV1 knockout and the same proliferation rate as wild-type Caco-2 cells (HTB-37, purchased from ATCC at cell passages of 50 to 60) was generated using CRISPR-Cas9 genome-editing technology [46]. The resuscitation, passage, and subculture of cells were carried out according to conditions recommended by ATCC. The number and the viability of the cells in suspension were measured using a cell viability detector. The suspensions were diluted proportionally with fresh DMEM culture medium (with 10% fetal bovine serum) at 5 × 10^5^ cells/mL and then inoculated into 6-well cell culture plates. Then, 2.5 mL of cell suspensions were inoculated per well and incubated at 37 °C and 5% CO_2_ for static culture. When cell growth reached 80–90% confluence, the cells were divided into six groups of 6-well cells each. Groups one and two were the control groups, which received fresh DMEM (The concentration of DMSO was the same as that of LPSC group). Groups three and four were the LPS groups, which received 1 μg/mL LPS in DMEM (the concentration of DMSO was the same as that of the LPSC group). Groups five and six were the LPSC groups, which received DMEM with 1 μg/mL LPS and 75 μM capsaicin (30.5 mg capsaicin dissolved in 1 mL DMSO solution, prepare a 0.1 M solution). The concentration of capsaicin was determined according to the Cell Counting Kit-8 (Tongren Chemistry Co., Ltd., Kumamoto, Japan), and 75 μM was the maximum concentration of capsaicin with the lowest cytotoxicity to Caco-2 (*TRPV1*^−/−^) cells. The cells were mixed gently and cultured at 37 °C and 5% CO_2_. After 24 h of culture, the medium was removed for later assay. The plates were placed on ice and rinsed with 2 mL of pre-cooled phosphate-buffered saline (PBS). After removing PBS, the first, third, and fifth groups were given 1 mL of pre-cooled RNA extraction reagent (TRI Reagent) to extract RNA for the quantitative determination of mRNA. The second, fourth, and sixth groups were treated with 200 μL pre-cooled SD001 lysate to extract protein for western blotting.

### 4.3. Detection of Biochemical Indicators

Serum triglycerides, blood glucose, cholesterol, low-density cholesterol, high-density cholesterol, and other biochemical indicators were determined using a BS-200 automatic biochemical analyzer (Shenzhen Mindray Medical, Shenzhen, China). Serum insulin concentrations were measured using a mouse insulin ELISA kit (Mercodia, Uppsala, Sweden).

### 4.4. Glucose Tolerance Tests

After 11 weeks of treatment, mice were fasted overnight (9 h). They were then gavaged with 1 g/kg body mass D-glucose solution, and their blood glucose concentrations were measured with a glucometer (ACCU-CHEK Aviva meter, Roche, Cupertino, CA, USA) at 0, 15, 30, 60, and 120 min.

### 4.5. Quantitative PCR Detection

We used 200 mg of fecal samples to extract total DNA. Briefly, after adding 20 μL RNase A (10 mg/mL), the samples were incubated at 37 °C for 20 min to remove RNA. The purity and concentration of the extracted total DNA were measured on a Nanodrop ND-2000 instrument. The standard of the purity determination is that the OD_260/280_ value should be between 1.8 and 2.0, and the concentration of the sample was diluted to 20 ng/μL. According to the primers, probes, reaction system, and conditions reported in the literature (Table 2), the contents of *Helicobacter*, *Desulfovibrio*, and *Sutterella* in the feces of mice in each group were measured using quantitative fluorescence PCR [47,48,49]. The changes in the relative amounts of Proteobacteria in the HFD and HFDC groups were measured with reference to the value of mice in the SLD group.

### 4.6. Detection of Endotoxin

The anticoagulated blood samples were centrifuged at 1000 rpm for 15 min, and the supernatants were removed to measure LPS concentrations. The method was carried out according to the instructions of the endotoxin test kit (Toxinsensor^TM^ chromogenic LAL endotoxin assay kit).

### 4.7. Detection of Inflammatory Cytokines

Serum samples from mice were collected as follows: anticoagulated blood samples with EDTA were centrifuged at 1000 rpm for 15 min, and the supernatants were removed for measurements.

The mice’s intestinal tissue was treated as follows: intestinal tissue was rinsed with pre-cooled PBS (0.01 M, pH = 7.4, containing protease inhibitor) to remove residual blood. Equal amounts of each intestinal tissue were mixed and minced. Minced tissue and PBS solution were placed in glass homogenizers according to the weight to volume ratio of 1:5; they were fully ground on ice and then subjected to ultrasonic crushing. Finally, homogenates were centrifuged at 5000 rpm for 10 min, and supernatants were removed for measurement.

Supernatants of culture medium were treated as follows: Cell culture medium was transferred to sterile centrifuge tubes and centrifuged at 1000 rpm at 4 °C for 20 min, and the supernatants were removed for measurement. According to the manufacturer’s instructions, levels of IL-6 and TNF-α in serum, intestinal tissue, and supernatants were measured using a double-antibody sandwich ELISA kit.

### 4.8. Histopathological Examination

Fresh duodenum, jejunum, and ileum were fixed in 4% paraformaldehyde for 24 h at room temperature. The fixed samples were processed, embedded in paraffin, sectioned at 2 to 4 μm, stained with hematoxylin and eosin, and examined under an optical microscope. Images were obtained from areas with the best quality. The villus length and recess depth of each intestinal segment were measured using image analysis software to calculate the villus gland ratios.

### 4.9. Detection of the Relative Expression Level of RNA

In each group, mice’s duodenum, jejunum, and ileum tissues were homogenized; 50 mg of tissue homogenates and cell samples were used for total RNA extraction. The cDNA obtained by reverse transcription was diluted using the two-fold dilution method. The standard curves for amplifying each gene were generated in our laboratory according to the primers, reaction system, and conditions reported in the literature (Table 3) to verify the amplification efficiency [12,14,50,51,52]. According to the instructions of the TB green ^®^ Premix Ex Taq™ II kit, quantitative detection of each gene was performed on the CFX96 Touch™ real-time PCR detection system, with *GAPDH* as the internal reference gene. Calibrations and normalizations were performed using the 2^−ΔΔCT^ method. The relative expression levels of genes in different groups were calculated to determine the relative expression levels of each gene.

### 4.10. Western Blotting

Total protein from the duodenum, jejunum, and ileum of mice in each group was extracted according to the operation instructions of Minute^TM^ Total Protein Extraction Kit for Animal Cultured Cell and Tissues. Total protein was extracted from 20 mg of tissue homogenates and cell samples. Protein concentrations were measured for western blotting. Antibody dilution ratios of target proteins ZO-1, occludin, and GAPDH were 1:2000, 1:1000, and 1:2500, respectively.

### 4.11. Measurement of Trans-Epithelial Electrical Resistance (TEER)

Cell suspensions were subjected to viability detection and counting using a cell viability detector. With fresh DMEM culture medium, the cells that survived at more than 95% were diluted into suspension at 2.5 × 10^5^ cells/mL, then inoculated into 12-well Transwell culture plates (pore size 0.4 μm). We added 1.5 mL fresh DMEM into the lower compartments and inoculated 0.5 mL of cell suspension into the upper compartments. Two wells were reserved for each plate as blank controls. We added 0.5 mL DMEM into the upper chamber and 1.5 mL fresh DMEM into the lower chamber. Finally, the plate was placed in an incubator at 37 °C and 5% CO_2_ for static culture.

The culture medium was replaced every other day during the first week and every day after one week. The resistance of each well and the cell membrane integrity were measured after 21 days. Wells with similar cell resistance values were selected and divided into three groups of six wells per group. The first group was the control group, to which was added with fresh DMEM, and the second group was the LPS group, to which was added with 1 μg/mL LPS in DMEM. The third group was the LPSC group, to which was added with DMEM with 1 μg/mL LPS and 75 μM capsaicin. After mixing gently, the cells were incubated at 37 °C with 5% CO_2_, and the transmembrane resistance of cells in each group was measured at 3, 6, 12, 18, and 24 h.

### 4.12. Statistical Analysis

Statistical analyses were conducted with GraphPad Prism 8.0 software (GraphPad Software, San Diego, CA, USA). After testing the normal distribution of sample data, one-way analysis of variance and Dunnett’s tests were used to compare the differences between groups. All data were expressed as mean ± SD, and a *p*-value less than 0.05 indicated statistically significant differences.

## 5. Conclusions

Capsaicin mediates lower weight gain by reducing CLGI. Capsaicin reduces the relative abundance of Proteobacteria in the intestine, reduces the concentration of LPS in the intestine, weakens the local inflammation mediated by TLR4, and thus reduces the damage to the intestinal barrier. Capsaicin promotes the upregulation of the expressions of transmembrane protein ZO-1 and cytoplasmic protein occludin, improving TEER. The compound restores intestinal barrier function and reduces intestinal permeability, thereby reducing the concentration of LPS entering the circulation and alleviating endotoxemia. This phenomenon reduces serum concentrations of inflammatory cytokines (TNF-α and IL-6), weakens CLGI, and ultimately reduces weight gain, food intake, levels of triglyceride and cholesterol, and fasting blood glucose levels. There is also improvement in glucose tolerance and insulin levels in HFDC-fed *TRPV1*^−/−^ mice, helping the animals lower weight gain. This study systematically elucidates the mechanism by which capsaicin reduces LPS concentration in the intestine, attenuates CLGI, and facilitates lipid reduction and weight loss. It was confirmed that capsaicin can stimulate the expression of the intestinal transmembrane protein ZO-1 and the cytoplasmic protein Occludin, increase TEER values, and repair intestinal barrier function. These findings further enhance our understanding of capsaicin’s lipid-lowering mechanisms. Capsaicin’s ability to repair the intestinal barrier function not only reduces the translocation of inflammatory factors into the bloodstream but also significantly affects lipid absorption. Therefore, further studies are necessary to determine whether capsaicin’s impact on intestinal lipid absorption is a key mechanism underlying its lipid-lowering effects.

## Figures and Tables

**Figure 1 ijms-25-08979-f001:**
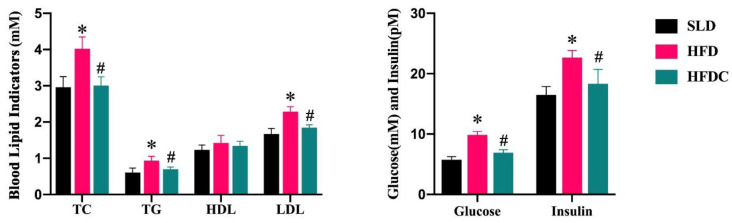
Effects of capsaicin on plasma parameters in *TRPV1*^−/−^ mice. Data are expressed as mean ± SD (*n* = 10). *, HFD group vs. SLD group, #, HFDC group vs. HFD group. (After testing the normal distribution of sample data, one-way ANOVA followed by Dunnett’s test was performed. *p* < 0.05.)

**Figure 2 ijms-25-08979-f002:**
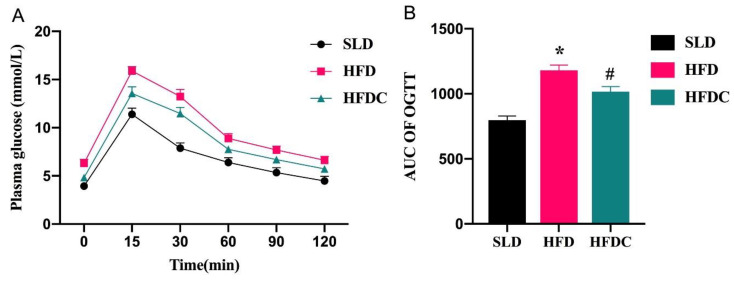
Effects of capsaicin on the oral glucose tolerance test (OGTT) of *TRPV1*^−/−^ mice. Data are expressed as mean ± SD (*n* = 10). *, HFD group vs. SLD group, #, HFDC group vs. HFD group. (After testing the normal distribution of sample data, one-way ANOVA followed by Dunnett’s test was performed. *p* < 0.05.)

**Figure 3 ijms-25-08979-f003:**
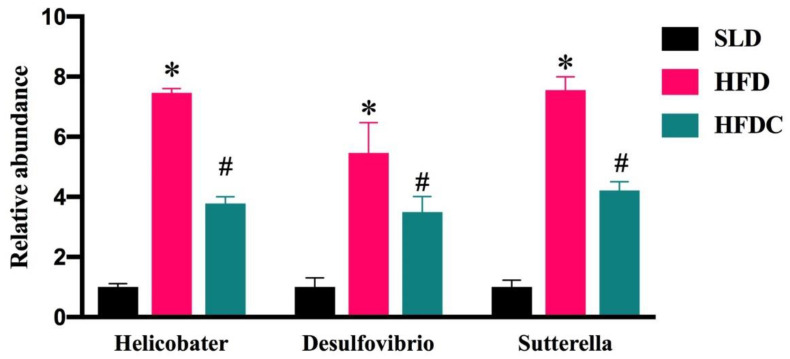
Capsaicin reversed the HFD-induced alteration in the microbial composition. Data are expressed as mean ± SD (*n* = 10). *, HFD group vs. SLD group, #, HFDC group vs. HFD group. (After testing the normal distribution of sample data, one-way ANOVA followed by Dunnett’s test was performed. *p* < 0.05.)

**Figure 4 ijms-25-08979-f004:**
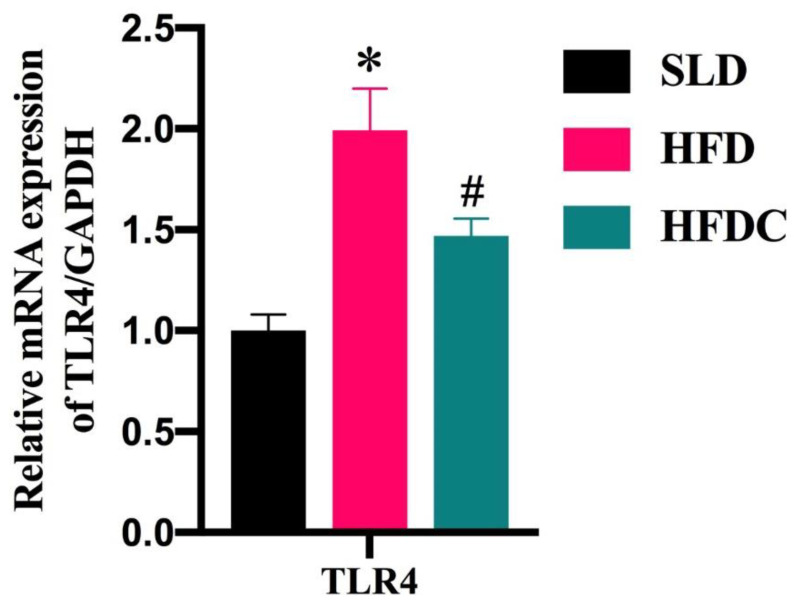
The expressions of *TLR4* gene in the intestinal tissue of *TRPV1*^−/−^ mice. Data are expressed as mean ± SD (*n* = 10). *, HFD group vs. SLD group, #, HFDC group vs. HFD group. (After testing the normal distribution of sample data, one-way ANOVA followed by Dunnett’s test was performed. *p* < 0.05.)

**Figure 5 ijms-25-08979-f005:**
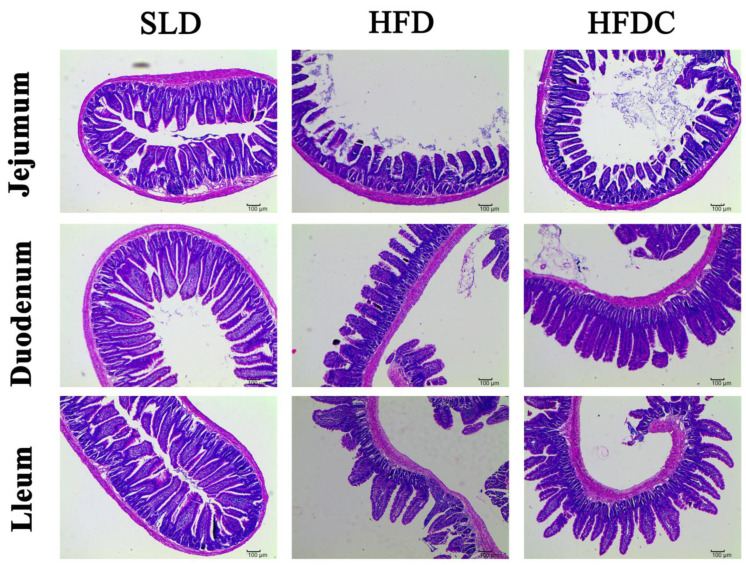
Effect of capsaicin on the histopathology of the intestinal tract in *TRPV1*^−/−^ mice.

**Figure 6 ijms-25-08979-f006:**
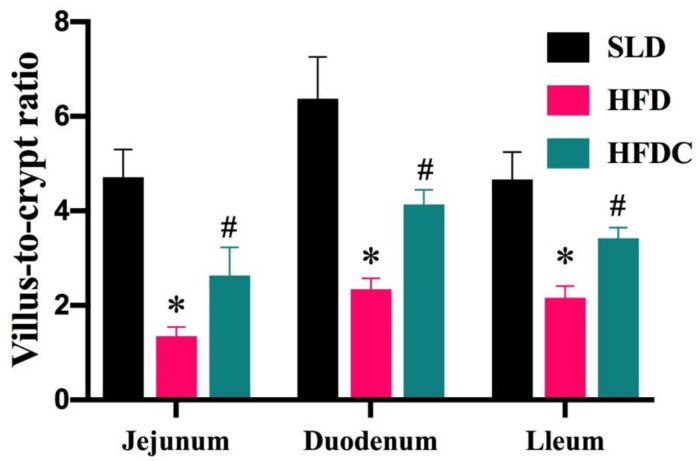
Effect of capsaicin on the villus-to-crypt ratio of the intestinal tract in *TRPV1*^−/−^ mice. Data are expressed as mean ± SD (*n* = 10). *, HFD group vs. SLD group, #, HFDC group vs. HFD group. (After testing the normal distribution of sample data, one-way ANOVA followed by Dunnett’s test was performed. *p* < 0.05.)

**Figure 7 ijms-25-08979-f007:**
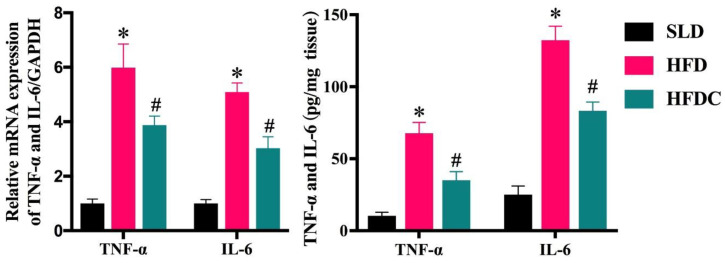
The expressions of *IL-6* and *TNF-α* gene in the intestinal tissue of *TRPV1*^−/−^ mice. Data are expressed as mean ± SD (*n* = 10). *, HFD group vs. SLD group, #, HFDC group vs. HFD group. (After testing the normal distribution of sample data, one-way ANOVA followed by Dunnett’s test was performed. *p* < 0.05.)

**Figure 8 ijms-25-08979-f008:**
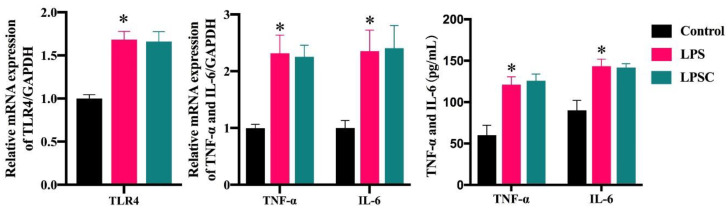
The expressions of *TLR4*, *IL-6* and *TNF-α* gene in Caco-2 (*TRPV1*^−/−^). Data are expressed as mean ± SD (*n* = 6). *, LPS group vs. control group. (After testing the normal distribution of sample data, one-way ANOVA followed by Dunnett’s test was performed. *p* < 0.05.)

**Figure 9 ijms-25-08979-f009:**
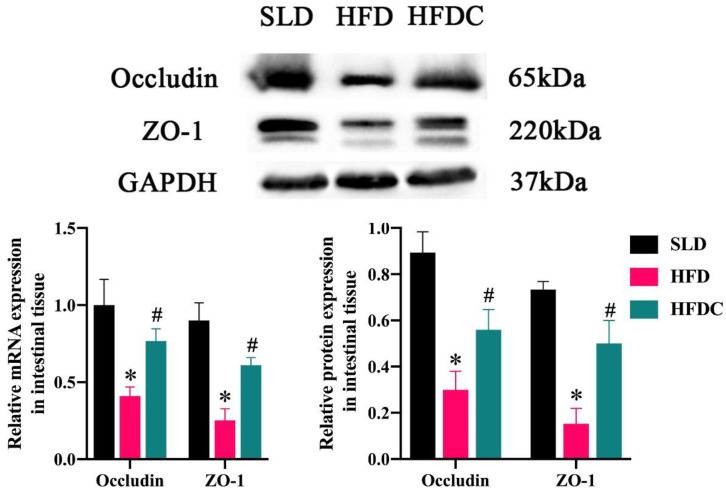
The expressions of occludin and ZO-1 in the intestinal tissue of mice. Data are expressed as mean ± SD (*n* = 10). *, HFD group vs. SLD group, #, HFDC group vs. HFD group. (After testing the normal distribution of sample data, one-way ANOVA followed by Dunnett’s test was performed. *p* < 0.05.)

**Figure 10 ijms-25-08979-f010:**
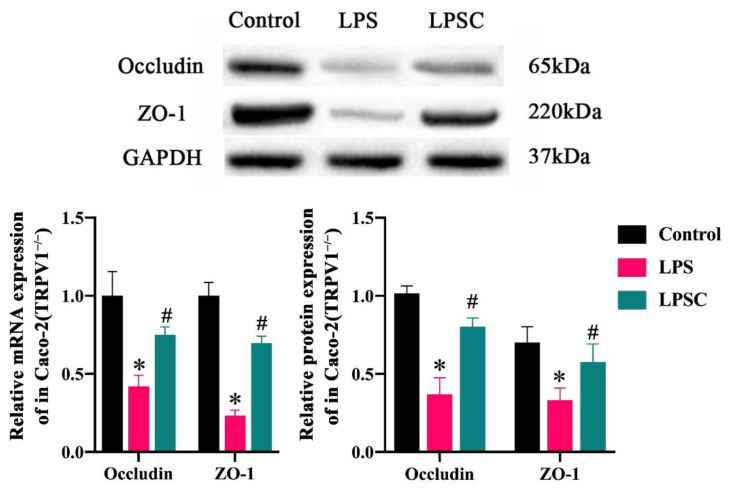
The expressions of occludin and ZO-1 in Caco-2(*TRPV1*^−/−^). Data are expressed as mean ± SD (*n* = 6). *, LPS group vs. Control group, #, LPSC group vs. LPS group. (After testing the normal distribution of sample data, one-way ANOVA followed by Dunnett’s test was performed. *p* < 0.05.)

**Figure 11 ijms-25-08979-f011:**
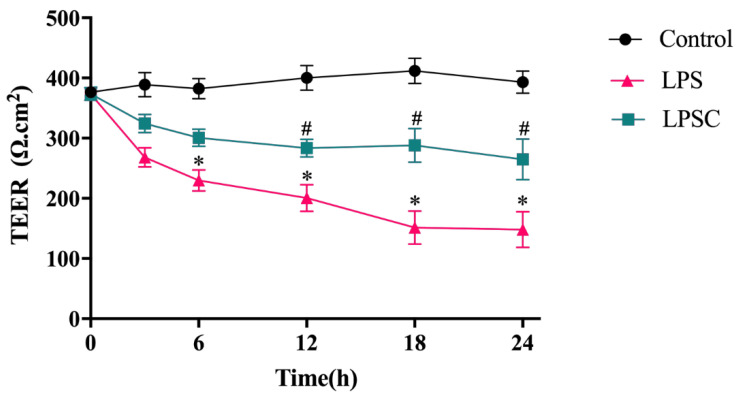
The influence of capsaicin on Caco-2 (*TRPV1*^−/−^) cells on TEER values. Data are expressed as mean ± SD (*n* = 6). *, LPS group vs. Control group, #, LPSC group vs. LPS group. (After testing the normal distribution of sample data, one-way ANOVA followed by Dunnett’s test was performed. *p* < 0.05.)

**Figure 12 ijms-25-08979-f012:**
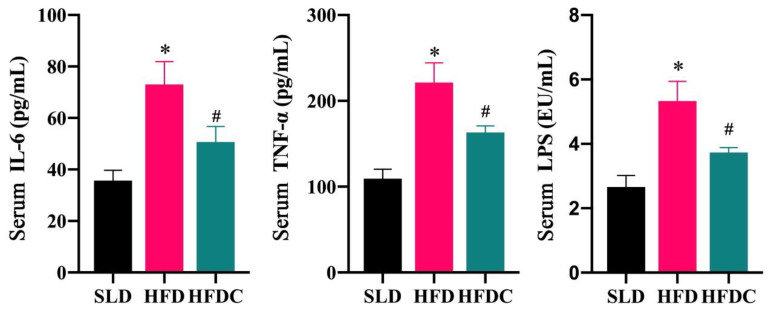
IL-6, TNF-α and LPS concentrations in the serum of *TRPV1*^−/−^ mice. Data are expressed as mean ± SD (*n* = 10). *, HFD group vs. SLD group, #, HFDC group vs. HFD group. (After testing the normal distribution of sample data, one-way ANOVA followed by Dunnett’s test was performed. *p* < 0.05.)

**Table 1 ijms-25-08979-t001:** Effects of capsaicin on food intake and body weight in *TRPV1*^−/−^ mice.

Parameter	SLD	HFD	HFDC
Initial Body Weight (g)	22.12 ± 0.95	21.90 ± 0.74	22.27 ± 1.06
Final Body Weight (g)	31.04 ± 0.96	41.06 ± 1.16 *	35.70 ± 1.65 ^#^
Body Weight Gain(g)	8.92 ± 0.67	19.16 ± 0.63 *	13.43 ± 1.52 ^#^
Average Feed Intake (g/d)	3.27 ± 0.13	5.16 ± 0.11 *	4.09 ± 0.13 ^#^
Total Feed Intake (g)	274.47 ± 11.34	433.16 ± 9.40 *	343.42 ± 10.80 ^#^

Data are expressed as mean ± SD (*n* = 10). *, HFD group vs. SLD group, #, HFDC group vs. HFD. (After testing the normal distribution of sample data, one-way ANOVA followed by Dunnett’s test was performed. *p* < 0.05.)

**Table 2 ijms-25-08979-t002:** List of primers of gut microbiota.

Bacterial Species	Primer	Sequence(5′-3′)	References
Helicobater	Forward	GCTCTCACTTCCATAGGCTATAATGTG	[47]
	Reverse	GCGCATGTCTTCGGTTAAAAA	
	Probe	FAM-TAGGGCCTATGCCTACCCCTGCGA-TAMARA	
Desulfovibrio	Forward	CCGTAGATATCTGGAGGAACATCAG	[48]
	Reverse	ACATCTAGCATCCATCGTTTACAGC	
Sutterella	Forward	CGCGAAAAACCTTACCTAGCC	[49]
	Reverse	GACGTGTGAGGCCCTAGCC	
	Probe	FAM-CACAGGTGCTGCATGGCTGTCGT-NFQ	

**Table 3 ijms-25-08979-t003:** The primers used for real-time RT-PCR.

Gene	Primer	Sequence(5′-3′)	References
ZO-1(human)	Forward	GGTGAAGTGAAGACAATG	[14]
	Reverse	GGTAATATGGTGAAGTTAGAG	
Occludin(human)	Forward	GAGTTGTATCTGTTGTTGT	
	Reverse	TTCGTGGTATAGCATTCT	
IL-6(human)	Forward	GACAGCCACTCACCTCTTCA	
	Reverse	TTCACCAGGCAAGTCTCCTC	
TNF-α(human)	Forward	GTCAGATCATCTTCTCGA ACC	
	Reverse	CAGATAGATGGGCTCATACC	
TLR 4(human)	Forward	CTGGAAATATGACCACAGTCAGAA	[50]
	Reverse	TCAATCA CCCTAGACCTGCTCAA	
GAPDH(human)	Forward	CTGACTTCAACAGCGACACC	
	Reverse	AGCCAAATTCGTTGTCATACC	
Occludin(mice)	Forward	CCTTCTGCTTCATCGCTTCCTTA	[51]
	Reverse	CGTCGGGTTCACTCCCATTAT	
ZO-1(mice)	Forward	GATAGTTTGGCAGCAAGAGATGGTA	
	Reverse	AGGTCAGGGACGTTCAGTAAGGTAG	
IL-6(mice)	Forward	CACATGTTCTCTGGGAAATCG	[52]
	Reverse	TTGTATCTCTGGAAGTTTCAGATTGTT	
TNF-α(mice)	Forward	GCCACCACGCTCTTCTGTCTAC	
	Reverse	GGGTCTGGGCCATAGAACTGAT	
TLR 4(mice)	Forward	CGCTTTCACCTCTGCCTTCACTACAG	[12]
	Reverse	ACACTACCACAATAACCTTCCGGCTC	
GAPDH(mice)	Forward	GCATCCACTGGTGCTGCC	
	Reverse	TCATCATACTTGGCAGGTTTC	

## Data Availability

This published article includes all data generated or analyzed during this study.

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
