# Peer review of "Capsaicin Reduces Obesity by Reducing Chronic Low-Grade Inflammation"

_ijms, 2024, doi:10.3390/ijms25168979_

Round 1
Reviewer 1 Report
Comments and Suggestions for Authors
Manuscript looks scientifically relevant and impactful. However, introduction section is poorly written. Methodology section should have proper citation regarding the protocol they have followed. I have following comments
1.Line 34- Please change the citation format in the text as per journal format.
2. Line 47-49- Authors should give in depth explanation on all possible contributing factors for high level of bacteria associated LPS in the human intestine.
3. Line 57-67- I suggest the authors to include this paragraph in your discussion section.
4. Line 75-81- It is irrelevant to provide the information about the methodology you adopted in introduction section.
5. In the introduction section authors are suggested to provide more literature about obesity, highlighting the possible cause and mechanism at molecular level. Similarly, information on curcumin given in the introduction section is very low. Authors could highlight the literature on about chemical structure, and possible pharmacological effect of capsiscin.
6. Line 256- Authors should explain more on how recruitment of inflammatory cytokine ultimately result in obesity.
7. In methodology section authors have not cited the suitable reference regarding the protocol they have followed. Also, I could not see the reference 41 to 46 cited in the text. Thus, please cite every protocol you followed with suitable references.
8. Figure 13 should be moved in discussion section
Comments on the Quality of English LanguageMinor improvement in english is needed
Reviewer 2 Report
Comments and Suggestions for Authors
Comments:
My main concern about this study is that the authors do not propose the mechanisms by which capsaicin acts in the absence of TRPV1. This is particularly important because there was a decrease in food intake. It means that the lower weight gain could, in itself, exert this effect: lower weight, lower inflammation, and lower changes in intestinal permeability. Virtually all in vivo results could simply be the result of lower weight gain. The results of Figure 8 reinforce the lack of effects on inflammatory cytokines in KO cells.
To address this issue, the authors should investigate possible mechanisms or maintain the food intake of the HFD group similar to that of capsaicin (mouse- matched intake).
The methods do not describe whether the authors verified the data normality since the ANOVA test requires a normally distributed sample population. Moreover, the authors used one-way ANOVA, but the comparison between the SLD group and the HFDC was not presented. This comparison is interesting because it reveals whether there was a total or partial recovery of the analyzed parameters in HFDC compared to the SLD.
What is the reason for using CaCo-2 cells, which are adenocarcinoma cell lines with a very different profile from that observed in KO cells?
Regarding intestinal bacteria, it is very challenging to draw any conclusions by analyzing only one phylum (proteobacteria). It is interesting to observe the balance among various phyla to conclude the changes in the microbiota. Again, this reduction may have been solely due to weight loss.
It is essential to state the limitations and the contribution (novelty) of your article to the knowledge on the subject.
In the conclusion, the authors report that capsaicin mediates weight loss by reducing inflammation. The authors do not have a basis in their results to conclude this. Capsaicin could mediate inflammation through reduced weight gain. (Remembering that there was no weight loss, only lower weight gain).
The conclusion the authors draw regarding capsaicin should be replaced with the HFDC group.
Minor comments:
- Why was capsaicin given on alternate days?
- Was the stomach not examined? It would be important to see if there are any irritative lesions related to the gavage of capsaicin that may have hindered food intake.
- Only two inflammatory cytokines were measured. It would be interesting to observe levels of others with a more anti-inflammatory profile, such as IL-10.
Reviewer 3 Report
Comments and Suggestions for Authors
The manuscript by Yang, Li, and Wang investigates the anti-obesity effects of capsaicin supplementation to decrease low-grade inflammation by reducing the abundance of gut LPS-producing Proteobacteria and improving epithelial barrier function in the intestine, thereby decreasing the ability of gut LPS to gain systemic access, which attenuates TLR4 activation and downstream cytokine production. They conduct appropriate in vitro cell culture and in vivo mouse studies and show that the effects of capsaicin are not through TRPV1. Generally, the studies are well-performed and the results are valuable and presented appropriately.
The authors should make clear that their work shows associations between capsaicin-mediated reductions in the high-fat diet-induced increases in Proteobacteria, but does not establish a cause-effect relationship between the bacteria and the low-grade inflammation. Simply tone down the language used to interpret these results.
Line 40 and throughout the manuscript - Capitalize the ‘G’ in Gram.
Line 84 – describe briefly the composition of the high-fat diet used in these studies
Lines 84-85 – define HFD and SLD upon their first use
Line 86 – add “obesity’ between ‘the’ and ‘model’
Line 88 – define HFDC upon its first use
Figure 1 – define the acronyms for the blood lipid indictors
Indicate why only TRPV1 knockout mice and cells were used for the studies.
\Whereas TLR4 is indeed a receptor for LPS, explain why Tlr4 expression was increased by HFD. By which cells is the Tlr4 expressed in intestinal tissue. Is this from leukocytes, epithelium, other cells?
Line 172 – change “TIL-6” to “IL-6”.
Line 179 – the rationale for the Caco-2 studies is not introduced and the experimental design is not briefly described. Instead, only the results are presented.
Line 185 – describe the “KO cell model”
Line 189 – define TEER and describe why it was measured in these studies
Figure 11 – reorder and recolor the groups to reflect their presentation in other figures (e.g., 8 and 10).
Figure 12 – The Y-axis labels should read “Plasma”, not “Plas”. In addition, the figure legend implies these samples are serum. Which was it? Line 371 implies that plasma was prepared and used in these studies. Be consistent throughout the manuscript.
Table 1 – is the probe sequence for Desulfovibrio missing or was this assessed with primers only?
Line 468 – TEER was trans-epithelial (not trans-endothelial) electrical resistance
Line 488 – the figures indicate that Dunnett’s tests were performed, whereas here it indicated Duncan tests were used. Which is it?
Figure 13 – the peppers imply capsaicin, but why not include the word in the figure to make the concept clearer?
Reviewer 4 Report
Comments and Suggestions for Authors
Dear authors,
The manuscript "Capsaicin Reduces Obesity by Reducing Chronic Low-Grade Inflammation" covers an interesting topic. Their previous studies have found that capsaicin exhibits anti-obesity effects without activating the TRPV1 cation channel. In this study, the authors aimed to determine the anti-obesity effects of capsaicin in a diet-induced obesity animal model using TRPV1 knockout mice and the intestinal epithelial cell line Caco-2. They observed alterations in the abundance of Proteobacteria and LPS levels, which were associated with lower body weight gain. They concluded that the anti-obesity effect of capsaicin is mediated by reducing chronic low-grade inflammation (CLGI) without TRPV1 cation channel activation, thereby controlling body weight gain. I find this study to be well-conducted with relevant results. The elucidation of the mechanism by which capsaicin mediates weight loss by reducing CLGI in vivo and in vitro is somewhat novel. The findings regarding capsaicin’s ability to repair intestinal barrier function and the explanation of its mechanism are important to publish. However, there are still some issues that need to be addressed and areas that need to be enhanced to improve the scientific quality of the manuscript.
Specific comments that need clarification and revision are as follows:
1. A very important finding about capsaicin's ability to repair intestinal barrier function and the explanation of its mechanism is missing in the abstract and conclusions.
2. The authors used female mice, which are subject to hormone variations during periodic cycles. How did they manage this possible bias in the results? Is this a limitation of the study?
3. No rationale for the dose of capsaicin is provided. Can the authors explain why gavage every other day for 12 weeks was considered an appropriate approach versus simple dietary incorporation?
4. SLD (D12450B from Research Diets) and HFD (D12492 from Research Diets) have different caloric densities. Please show daily food intake by calories rather than grams when presenting all three groups in one figure to prove that the DIO model was established by excess calorie intake and that capsaicin lowered food intake and body weight gain in both SLD and HFD groups at similar or different rates.
5. Is the same antibody used when determining the expression of ZO-1 and occludin in intestinal tissues of mice and in Caco-2 cells? Is the dilution the same?
6. Blood glucose levels in Figures 1 and 2 are inconsistent. Although there are several differences such as measuring time points, fasting time, routes of blood collection, and measurement methods, the fasting glucose levels in the HFD groups differ significantly. Please check the raw data again and provide reasonable explanations.
7. The citation of several references is confusing in the manuscript. For example, in Line 35, Line 253, Line 258, and Line 304, the sentences should include a reference.
Comments on the Quality of English LanguageA very important finding about capsaicin's ability to repair intestinal barrier function and the explanation of its mechanism is missing in the abstract and conclusions.
Reviewer 5 Report
Comments and Suggestions for Authors
In this paper, the authors provided valuable insight into the effect of capsaicin on obesity. This paper is interesting and well-written. However, the following changes are required to improve the paper:
Line 18 – the sentence is not clear, please correct the following sentence in the Abstract: „LPS receptor toll-like receptor 4(TLR4) levels decreased following decreased of LPS levels.“
The Methods and Materials section should be relocated to precede the Results section. This will allow readers to gain an insight into the methodologies before reviewing the results.
Also, to evaluate the effect of capsaicin generally, an additional group receiving capsaicin alone is required in the experiments (SLD + capsaicin group). This would allow readers to determine whether capsaicin alone influences the experimental parameters or if its effects are specific to the context of HFD and obesity.
Another important thing is that the dissolution medium was not mentioned for LPS and capsaicin in the in vitro part. Was DMEM used to dissolve LPS and/or capsaicin or were they dissolved in another solvent prior to the experiment? If another solvent is used, an additional group containing the solvent should be added, in addition to the DMEM-only group.
Comments on the Quality of English LanguageThe English language is generally acceptable, however, minor editing is needed.
Round 2
Reviewer 2 Report
Comments and Suggestions for Authors
Here's the translation:
---
The authors improved the manuscript and made modifications that addressed my comments. However, the authors claim that they have shown for the first time that capsaicin induces ZO-1, which was recently demonstrated in the study of Santos et al. (https://pubs.acs.org/doi/10.1021/acsptsci.4c00207). Therefore, the authors should remove this statement from the conclusion or discuss Santos' results to support their conclusion.
Author Response
Dear Reviewer:
Thank you for taking the time to review our manuscript again. We are grateful for your constructive suggestions and criticisms, and we have revised the manuscript accordingly.
Comments 1: The authors improved the manuscript and made modifications that addressed my comments. However, the authors claim that they have shown for the first time that capsaicin induces ZO-1, which was recently demonstrated in the study of Santos et al. (https://pubs.acs.org/doi/10.1021/acsptsci.4c00207). Therefore, the authors should remove this statement from the conclusion or discuss Santos' results to support their conclusion.
Response 1: Thank you for giving us the affirmation of the article revision and insightful feedback on our research. We have rewritten the major findings and removed the phrase 'for the first time' from the Abstract and Conclusions sections on lines 29 and 539 of the revised manuscript. We have also incorporated a discussion of Santos's results into the discussion section on line 346, which supports their research conclusions.
Reviewer 5 Report
Comments and Suggestions for Authors
No additional changes required.
Author Response
Thank you for taking time out of your busy schedule to review our manuscript again.
Comments 1: No additional changes required.
Response 1: Thank you for your affirmation of our research.